# Recent Advances in Alkali-Activated Materials with Seawater and Sea Sand

**DOI:** 10.3390/ma16093571

**Published:** 2023-05-06

**Authors:** Zengqing Sun, Xiaoyu Li, Qingsong Liu, Qingyu Tang, Xiaochen Lin, Xiaohui Fan, Xiaoxian Huang, Min Gan, Xuling Chen, Zhiyun Ji

**Affiliations:** 1School of Minerals Processing and Bioengineering, Central South University, Changsha 410083, China; 2Nanjing Institute of Environmental Sciences, Ministry of Ecology and Environment of the People’s Republic of China, Nanjing 210042, China

**Keywords:** alkali-activated materials, seawater, sea sand, mechanical property, corrosion

## Abstract

The development of sustainable cementitious materials is essential and urgent for the construction industry. Benefiting from excellent engineering properties and a reduced greenhouse gas footprint, alkali-activated materials (AAM) are among the robust alternatives to Portland cement for civil infrastructure. Meanwhile, concrete production also accounts for around 20% of all industrial water consumption, and the global freshwater shortage is increasing. This review discusses recent investigations on seawater-mixed AAMs, including the effects of seawater on workability, reaction mechanism, shrinkage, short and long-term strength, binding of chloride and corrosion of steel reinforcement. Attention is also paid to the utilization of sea sand as aggregate, as well as discussions on the challenges and further research perspectives on the field application of AAMs with seawater and sea sand.

## 1. Introduction

Concrete, the key building material for infrastructure construction, contributes to improving quality of life and modern civilization. Worldwide concrete consumption has grown to over 10 billion tons per year, and the associated environmental and ecological problems are severe. To produce one ton of ordinary Portland cement (OPC), 0.8–1.0 ton of CO_2_ is released. The OPC industry is responsible for ca. 8% of global CO_2_ commission [1,2,3]. The urgency to develop and commercialize alternative construction materials with low CO_2_ properties is growing. Benefiting from excellent mechanical properties, acid and thermal resistance, strong adhesion to metallic and nonmetallic surfaces, reduced greenhouse gas footprint, etc., alkali-activated materials (AAMs) are robust candidate attracting attention from academic and industrial fields [2,3,4,5,6].

Research on AAMs can be dated back to the 1900s, when Kühl patented mixtures of slag with alkaline components, providing performance “fully equal to the best Portland cements” as a “developing material” [7]. In the 1940s, Purdon conducted extensive work on alkali-activated slag [8], and some of the findings were commercialized later in the Belgium [9]. Around the same period, Glukhovsky from the former Soviet Union developed AAMs using low-calcium or calcium-free aluminosilicates and utilized these in infrastructure constructions [10]. Several aluminosilicate-based AAM formulations were patented and termed as geopolymer by Davidovits in the 1970s [11,12], and the terminology “geopolymer” was widely accepted and used afterwards. Nowadays, AAMs have been investigated globally.

According to Provis and van Deventer [2,5], AAMs are normally classified into two subsets, i.e., high-calcium system (such as alkali-activated slag, AAS) and low-calcium system (geopolymer). The main reaction product of the former is calcium (alumino)silicate hydrate (C-A-S-H) gel, with a layered double-hydroxide group almost always coexisting as a secondary product. The C-A-S-H gel has a disordered tobermorite-like structure and can be regarded as aluminum-substituted C-S-H. C-A-S-H is structured of layers of tetrahedrally coordinated silicate chains, with Ca^2+^, alkalis and water located in the interlayer region [13,14,15,16,17,18]. In contrast, the fundamental binding phase in the geopolymer system is an alkali aluminosilicate gel, which is frequently represented as N-A-S-(H). Depending on the mixture composition, substitution of sodium by potassium and/or calcium can take place, resulting in a more complete description of the gel as N,K-(C)-A-S-(H). Both aluminum and silicon are present in tetrahedra coordination in the gels, with alkali cations balancing the negative charge caused by substitution of Si^4+^ by Al^3+^ [12,19,20,21].

Another concern about concrete production is the consumption of huge amounts of freshwater, which accounts for around 20% of all industrial water consumption [22]. However, two-thirds of the world population will suffer from water shortage stress by 2025, and the population in regions of water scarcity will be 1.8 billion [23]. As forecasted, 75% of water for concrete production will probably be consumed in regions of water shortage [22]. Consequently, adopting seawater as mixing and batching water for construction material production can not only reduce the short supply of freshwater, but also provide convenience for marine and offshore construction. Pilot research in this area has been conducted recently, with significant and positive findings. In this work, recent advances on AAMs produced using seawater were revisited and outlined. Workability, mechanical properties, drying shrinkage, microstructure, reaction production and chloride binding capacity, and durability, including acid/sulfate resistance and reinforcement corrosion, were addressed. Attention is also paid to AAMs containing sea sand and coral sand. This review is concluded by highlighting challenges, limitations, and opportunities for further research.

## 2. Properties of Fresh and Hardened AAMs with Seawater and Sea Sand

### 2.1. Workability

The workability of AAMs with seawater and sea sand has been described by several researchers (Table 1). No obvious differences between the setting time of tap water- and seawater-mixed AAM mortars have been reported by Shi et al. [24], in whose study the final setting time of seawater-mixed AAM mortar was the same as its tap water-mixed counterpart. A close check of the data showed that the initial setting time of sea water-mixed AAM mortar was 60 min, which was 5 min shorter than those mixed using tap water. A decrease of both the initial and finial setting time was detected in [25]. The setting time reduction was assigned to the acceleration of alkali activation caused by CaCl_2_, which was formed by reaction of Cl^−^ initially contained by seawater with dissolved Ca^2+^ (Table 2). Hydration accelerated by CaCl_2_ has been frequently reported in sea water-mixed cement systems, with a setting time reduction of up to 30%. In comparison, the reported setting time decrease caused by seawater mixing was less significant in AAM than cement systems. Meanwhile, a setting time increase was reported in [26]. A more significant increase was observed when using water with higher salinity (Reverse Osmosis (RO) reject water, with a salt concentration 2.5-times that of seawater). The authors explained that the salting-out effect (solubility reduction of sodium silicate due to interactions between salt ions with water in high-salinity water) might be responsible for the setting time increase [27].

In addition, sea sand possesses a significant influence on setting time and slump of AAMs. Compared with AAM mortar prepared with freshwater and river sand, utilization of seawater and coral sand caused an obvious decrease in setting times and slump [28]. The sudden drop in slump using coral sand in [29] was related to the higher porosity and water adsorption properties of coral sand when compared to river sand and sea sand, resulting in the decrease of active water-to-binder ratio. Meanwhile, replacing coral sand with sea sand resulted in a profound increase in setting times and slump; the values were even higher than those of the freshwater river sand counterpart [28]. It is hard to obtain a general conclusion about the effect of sand type on setting time, due to the incomplete experimental design; the phenomena might indicate that the workability of AAMs is complicated and determined by many factors.

Zhang et al. [29] stated that the most profound factors influencing setting time and slump were Na_2_O-to-binder ratio and water-to-binder ratio, respectively. An increase in reactive Na_2_O content results in an increase in the alkalinity of the reaction system, leading to the repaid dissolution of solid precursors and consequently accelerating the reaction kinetics and hardening processes [29]. Similarly to freshwater-mixed AAMs, an increase in activator modulus (molar ratio of SiO_2_/Na_2_O) can also accelerate the reaction process of seawater-mixed AAMs. This is more significant in high-calcium AAM systems, due to the repaid formation of C-A-S-H gels via the reaction of readily available silicate ions with dissolved calcium ions [29].

### 2.2. Mechanical Properties

The effects of seawater and sea sand on the mechanical properties of AAMs are the most extensively investigated. Strength enhancement has been frequently reported (Figure 1). In [41], the strength improvement at different curing stages varies. The compressive strength of seawater-mixed AAM samples were 16.6%, 11.4% and 6.2% higher than freshwater-mixed samples at 7 days, 14 days, and 28 days, respectively. Strength improvement seems to decrease with the extension of curing time, but long-term mechanical behaviors were missing, preventing establishment of a logically sound conclusion. Although comparable flexural and compressive strength was detected by Lv et al. [25] over a timescale of 1 year, the seawater-mixed mortar achieved slightly higher early strength than freshwater-mixed specimens. As aforementioned, seawater initially contains high amounts of salt, which can accelerate the alkali activation process. The repaid generated reaction products, together with salt crystallizations, can fill up the voids and consequently improves the mechanical properties of obtained mortars, especially at the early stages. Seawater exhibits higher pH values and volumetric mass density than freshwater, which might also contribute to the strength improvement [41]. Although a higher strength increase was obtained at 28 days, rather than 7 days, in [42], the relatively slower reactive property of fly ash might be responsible.

A strength decrease induced by seawater mixing was also reported [40]. Compared with specimens prepared with deionized water, the seawater-mixed mortars showeda compressive strength reduction of 30% at 7 days. However, comparable and even higher strength was observed at 28 days and 91 days, respectively. The changes in reaction process and the difference in reaction products were responsible, and will be discussed in the following sections. The influence of seawater type was characterized via sampling seawater from three different locations in [40], and the difference in terms of mechanical properties were negligible.

In general, full replacement of river sand by sea sand resulted in the decrease of mechanical properties, and the degree of reduction depends on the mix design of the binder and the physical properties of sea sand. Shinde and Kadam [43] reported that the 28-day compressive strength of fly ash geopolymer mortars activated using 8, 10 and 12 M NaOH decreased by 9.34%, 16.5% and 19.28%, respectively, when replacing river sand with sea sand. In a study conducted by Anbarasan and Soundarapandian [44], AAM was prepared using the mixture of FA and GGBS (mass ratio = 7:3) as the precursor and a sodium silicate solution containing NaOH as the activator. When replacing river sand with sea sand, the 7- and 28-day compressive strength decreased by 8.23% and 6.64%, respectively. Moreover, there are larger amounts of harmful substances (such as salts, seashell particles (CaCO_3_)) on the surface of sea sand than river sand. These materials, together with the surface texture, weakens the binding and interlocking between paste and aggregate [45,46]. Strength enhancement was reported by Nguyen et al. [47]: samples containing river sand exhibited up to 16% higher compressive strength and up to 10% splitting tension strength when compared to sea sand counterparts. The only difference in the mixture is sand fineness modulus, which is 1.7 for sea sand and 2.54 for river sand. Additionally, the sea sand is physically more rounded or cubical in shape. This might possess an influence on mechanical behavior, since OPC concrete strength increases with the increase of fine aggregate fineness modulus, due to the quantity of paste per surface area of aggregate increasing. Replacing river sand with sea sand affects the mechanical properties of AAM concrete significantly. In [48], river sand was gradually replaced by sea sand, and the compressive strength decreased accordingly. Decreases in compressive strength by 28%, 12.53% and 8.5% were observed at a replacement ratio of 25% at 7, 28, and 120 days, further increasing to 55.83%, 38.56% and 33.12% when fully replacing river sand with sea sand [48]. In contrast, replacing sea sand with coral contributes to the increase of compressive strength [28]. Aside from the coarse surface of coral sand promoting better interlocking, the porous characteristics of coral sand can absorb water from paste and reduce the local water-to-binder ratio at the paste–aggregate interface. The absorbed water can be regarded as temporarily stored by coral sand, which will be released during the hardening stage and afterwards, forming an internal curing effect [49].

Similarly to OPC samples, the compressive strength of AAMs with sea sand decreases with the increase of sand-to-binder ratio. As reported in [48], strength reductions of 13.8% and 27%, from 41.32 MPa to 35.61 MPa and 30.18 MPa, were observed as the mass ratio of sand-to-binder increased from 3.5 to 4 and 4.5, respectively. In the case of low sand-to-binder ratio, the pastes can fully cover the aggregates and fill the voids inside the specimens.

### 2.3. Drying Shrinkage

The drying shrinkage of AAMs, mainly originating from the evaporation of internal free water from pores of hardened concretes, exhibited similar tendencies as OPC samples, but with higher amplitudes (Figure 2) [28,29,50,51,52]. The main development of drying shrinkage took place at an early stage, especially the first 28 days. A delayed increase was separately observed in [28,50]. In comparison with samples containing river sand, the sea sand counterparts showed higher drying shrinkage. As characterized by Yang et al. [50], sea sand possessed higher void volume fraction; thus, the void size among particles might be higher. During manufacture, voids among aggregates would be filled by pastes, which determines the drying shrinkage characteristics of obtained concrete. Moreover, the higher density of sea sand when compared to river sand, as well as seawater over freshwater, might also matter. The constitutes of aggregate were normally replaced by weight in these studies, while mass ratios were unchanged. The amount of paste is then increased per unit volume in samples containing sea sand. However, the adoption of coral sand contributed to the decrease of drying shrinkage because of the water-storing property of coral sand, as aforementioned [28]. As can be inferred from Figure 2, the mix proportion of paste possessed obvious influence on the drying shrinkage behavior of AAMs. Increasing sodium silicate solution modulus from 1.0 to 1.6 led to the increase of drying shrinkage by 78.9% at 120 days.

### 2.4. Thermal Properties

The thermal properties of seawater-mixed AAMs were extensively investigated by Li et al. [38]. Mortar samples were heated from room temperature to 100, 200, 400, 600, 800 or 1000 °C, with a heating rate of 5 °C/min, and maintained at target temperatures for 30 min, followed by natural cooling down within furnace to room temperature, during which the thermal strain was documented. Mass and strength loss, surface and microstructure change were also characterized. Gradual mass loss was observed with the increase of heating temperature, due to the loss of free water and/or bound water (Figure 3) [53]. Maximum mass loss, close to the water content in mixture, was obtained in the temperature range of 600–1000 ℃. The mass loss behaviors of seawater-mixed AAMs are very similar to the freshwater-mixed samples, excepting the minor variation at 100 ℃. The smaller mass loss of AAMs produced using seawater could be related to the presence of salts in seawater, which can retard water evaporation.

The vapor pressure caused by water evaporation, together with the temperature gradient, caused cracks on AAMs upon heating at temperatures ≥ 200 °C. The cracks were considerably more observed in pastes, rather than mortars or concretes. Cracks generated in AAM concretes were along the interfaces of paste with coarse aggregate, suggesting thermal incompatibility between AAM paste and aggregate.

Figure 4 compared the expansion or contraction of seawater-mixed AAM paste and concrete during heating. For paste, mass loss induced shrinkage of the specimens, while heating caused expansion; the balance of the contrary effect resulted in stable dimensions up to 150 °C. Further heating at 150–450 °C caused rapid contraction of AAM pastes, due to shrinkage of pores and dihydroxylation of reaction products caused by the loss of physically and chemically bonded water [54,55]. Afterwards, the dimension of tested paste specimens remained invariable at 400–600 °C because of the completion of water loss-assisted shrinkage. Considerable change was observed at 600–750 °C, which can be assigned to the softening and subsequent densification of AAM paste [56]. The slight expansion and contraction during the cooling process can be attributed to the shrinkage of solids upon cooling. Due to the pore structure being altered after heating, as well as water loss, most of the shrinkage is then irreversible. The more significant contraction of AAM paste when compared to OPC might be related to differences in the reaction mechanism and chemistry of reaction products.

When aggregates (sea sand as fine aggregate and basalt as coarse aggregate) were incorporated, samples exhibited an expansion during heating. Similarly, Kong and Sanjayan [57] reported thermal expansion of FA geopolymer concrete containing natural aggregates. Occupying weight ratio by ca. 80%, the thermal behavior of concrete is mainly determined by aggregates. More significant expansion characteristics of sea sand over river sand was inferred by Li et al. [38], for which the swelling of impurities (such as shell debris) on sea sand might also matter.

After thermal treatment, the strength decreased obviously for both seawater- and freshwater-mixed AAM paste (Figure 4, right). However, the strength drop of seawater-based samples was more significant, in which only 22% of the reference strength remained after exposure at 200 °C [38]. The sudden drop can be related to the cracks caused by water evaporation. Meanwhile, the residual strength of seawater AAM paste after 100 °C thermal treatment was higher than its freshwater counterpart. This is consistent with the smaller mass loss of seawater AAM paste in this temperature region. Samples mixed using seawater and sea sand achieved comparable residual strength as the one based on freshwater and river sand. The strength of AAM concretes depends on paste proportion, water-to-binder ratio, alkali content, curing conditions, aggregate gradation, etc. As aforementioned, concretes suffer from shrinkage caused by water evaporation, expansion of aggregates, and sintering of pastes. The residual strength is thus the combined results of these factors.

### 2.5. Reinforced and Confined AAMs

The bond strength, bond stiffness and failure modes of fiber-reinforced polymer (FRP)-reinforced AAM concrete with seawater and sea sand were characterized in terms of pull-out tests (Figure 5). According to Zhang et al. [58], the splitting failure of tested concretes took place mainly in the interfacial transition zone, and breaking of aggregates also occurred. In comparison with OPC concrete of the same strength grade, more pronounced aggregate breaking was detected in AAM concrete. The AAM concrete achieved a higher compressive-strength-to-splitting-tensile-strength ratio than its OPC counterpart, suggesting AAM contributed to improvement of the interfacial transition zone and strengthened the splitting tension strength of obtained concrete. In terms of bond behavior of FRP in AAM concrete with seawater and sea sand, both bar pull-out and cracking or rupture failure modes were detected. The pull-out mode was mainly observed in concrete of C40 grade. In contrast, cracking or rupture was the predominant failure mode in concrete exceeding 50 MPa. The FRP behaved as sudden crack or rupture at the load-end, when the applied load attained ultimate load, and the slippage of the free-end was relatively small. In both cases, abrasion and exfoliation on bar surfaces appeared without cracks, indicating that the AAM concrete possessed superior shear resistance to FRPs.

Benefiting from higher splitting tensile strength and a denser interfacial transition zone, the bond stiffness of FRP in AAM concrete, as well as the initial bond rigidity, is higher than its OPC counterpart. Concrete of high splitting tensile strength contributes to improvement of the constraining force and retards the generation of internal cracks. Moreover, both the bond stiffness and bond strength increase with the enhancement of the concrete strength grade, which can be related to the improved chemical adhesion and mechanical interaction between FRP and the surrounding concrete. Due to the weakness characteristics, including inadequate shear resistance and elastic modulus, and material anisotropy of FRP, the spalling of surface rib/fiber of FRP was the key factor causing bond failure. When the failure mode is dominated by cracking or rupture of FRP, the development of bond stress during pull-out tests was limited [58].

Concrete-filled tubes possess high load-carrying capacity and good seismic performance, principally furnished by the confining effect of encasing the tube in concrete, and have been widely used in infrastructure construction. As shown in Figure 6, the properties of AAM concrete with seawater and sea sand-filled tubes depend on tube characteristics and filling types (e.g., fully filled, double-skin filled) [59]. Generally, higher ductility was achieved when a stainless-steel tube was used. Similar to OPC concrete-filled tubes, stainless steel-based SSACFT exhibited an outward folding failure pattern, i.e., local buckling [60,61]. Compared to a GFRP counterpart, the confinement of a stainless-steel tube endures under larger axial strain. GFRP-based filled tubes displayed the failure mode of tube rupture in a hoop direction. Buckling in a longitudinal direction was observed prior to reaching the ultimate load, and the inner tube buckled earlier than the outer. More than once, the buckling of GFRP tubes in a longitudinal direction was detected during the loading process.

## 3. Reaction Products and Microstructure of AAMs with Seawater and Sea Sand

### 3.1. Reaction Products

The reaction products of seawater-mixed AAMs can be classified into main reaction products and secondary products. C-A-S-H and N-A-S-(H) gels are the predominant products in conventional AAMs, which are also frequently detected in seawater-mixed AAMs. To date, a negligible difference in gel chemistry has been reported after adopting seawater. Numerous research papers and several reviews have discussed the gels and corresponding reaction processes in depth [4,21], and will not be repeated here. Attention will be paid to the formation of secondary products, in particular the chloride binding phases.

Surface adsorption and formation of Cl-bearing phases (such as Friedel’s salts, Cl-hydrocalumite and Cl-hydrotalcite) are the main binding mechanism of chloride in a cementitious system [62]. The chemical binding process via formation of Cl-bearing phases has been demonstrated as more effective in chloride uptake than the surface adsorption process [63]. In [1], a chloride bond content of 91% was measured in seawater-mixed AAS, while the value was 69% when fly ash was adopted as a solid precursor. The finding implied that slag-based AAMs were more effective in the uptaking and binding of chloride, which can be attributed to their considerable magnesia content, contributing to the formation of hydrotalcite [35,64,65,66]. Hydrotalcite is a typical layered double hydroxide (LDH) and possesses the ability of adsorbing free chloride ions from its surroundings. Chloride-bearing hydrotalcite and hydrocalumite were actually detected in a slag-based AAM in [1]. The difference between Friedel’s salt and other chloride-bearing LDHs was determined by Khan et al. [1]. Given that there are great similarities in structure between Friedel’s salt and Cl-hydrocalumite, the XRD patterns are not the same and cannot be taken as an identical phase [67]. The high calcium content in slag can also promote the formation of C-A-S-H gel. Several individual investigations have reported the physical adsorption of chlorides on C-S-H [68,69,70]. Physical uptake of chloride by C-A-S-H has also been proposed [68,71]. The C-A-S-H theoretically presents a positively charged surface in alkaline conditions [72], which will most likely contribute to the uptake of Cl^−^. Decreasing Ca/Si ratio resulted in the reduced binding capacity of C-A-S-H on Cl^−^ because of the less positive surface charge [73]. In comparison, although chloride-bearing zeolites (chabazite and sodalite) were formed in seawater-mixed fly ash geopolymer, the content of these zeolites were limited [71,74]. Moreover, the chloride binding capacity of these zeolites deserves further investigation.

The alkalinity and chloride availability of AAMs systems also affects the chloride binding performance, as high alkalinity contributes to enhancing the reaction degree and, consequently, generating more C-A-S-H [75], which is beneficial for the adsorption of chloride ions. Meanwhile, because of the competing adsorption between Cl^−^ and OH^−^, the amount of chloride binding was reported to decrease with the increase of OH^−^ concentration in pore solution [72,76,77]. A reduced OH^−^/Cl^−^ ratio leads to decreased surface charge density and a thinner diffuse layer; thus, the chloride retained in the diffuse layer might be dropped [72]. Chloride availability determines the formation of chloride-bearing products. In [40], more chloride-bearing phases were formed when seawater with initial higher chloride content was used. In this case, the replacement of carbonate anions by chloride ions was observed in terms of the XRD pattern. The replaced carbonate ions resulted in the formation of CaCO_3_. Other chloride-bearing phases, including AlClO and aluminum chloride hydrate (AlCl_3_(H_2_O)_6_), were also detected.

### 3.2. Microstructure

The microstructure and morphology of AAMs are normally characterized using SEM (Figure 7), coupled with EDS in some cases to qualitatively analyze the constitution of reaction products. In SEM images, unreacted precursor particles and newly formed gels are integrated together at an early stage. Since the precursor particles were partially activated at an early stage, unconsolidated reaction products depositing on the surface of unreacted particles was frequently observed. The number of unreacted particles decreased with the aging, and gels were gradually densified, forming denser microstructures [28]. Comparing with conventional AAMs, the seawater-mixed samples showed looser microstructures at very early stages, but comparable or more compact layouts at later stages [24,78], implying that seawater might inhibit alkali activation at early stages while accelerating reactions at later stages. In addition, the modification effect of salt on pores and the interaction between salt with alkaline activator-generating nano-sized particles might also be responsible [24,35,79,80,81]. An ion exchange reaction between seawater and an activator solution cannot be avoided during alkali activation, converting soluble ions into insoluble particles. In [24], the precipitation of nano-sized particles was directly observed. The particles bonded together and formed a relatively dense matrix, with the composition of Si-Mg-O-Ca-Na being 11.8%-3.0%-31.9%-0.9%-0.6% in terms of mass ratio. Shi et al. [24] hypothesized the reaction regarding the reaction products as amorphous SiO_2_ and M-S-H gels. These precipitations can contribute to refine the voids and can be beneficial for strength development. It should be mentioned that the pH drop due to ion exchange reaction retards the alkali activation, resulting in strength loss [24]. Increasing the salt concentration contributed to the denser microstructure of resulting AAMs [78]. This is further supported by the porosity evolution characterized using MIP. Pore size refinement was reported in [1], as a decrease in large voids and increase of small pores was detected from both AAS and fly ash geopolymer mixed using seawater.

The microstructure of AAM using sea sand as a fine aggregate was not as dense as that incorporating river sand. More empty spaces were observed in samples containing sea sand, which caused the reduced compressive strength of the obtained matrix when compared to the control, although the binder was of the same mix proportion [47]. Nguyen et al. stated that the fine particles of sea sand create obstacles for fully developing the AAM structure [47]. Meanwhile, coral sand was found to modify the interfacial transition zone (ITZ) because of its porous characteristics. Reaction products can penetrate into the inner spaces of coral sand, facilitating the formation of compact and dense interactions between paste [47].

## 4. Durability of AAMs with Seawater and Sea Sand

As aforementioned, the adoption of seawater and sea sand in cementitious materials may lead to problems in long-term safety and stability. The durability behaviors of AAMs with seawater and sea sand are discussed in acid and sulfate resistance, and reinforcement corrosion.

### 4.1. Acid and Sulfate Resistance

After exposing the seawater AAM paste in 3% H_2_SO_4_ solution, gains in mass and compressive strength were observed at 7 days, followed by mass and strength loss [37]. The mass gain was related to the formation of sufoaminates. Newly formed phases can fill the voids and pores, which contribute to the strength increase. Extending the exposure time, the acid corrosion induced scaling and deterioration of AAM paste led to mass loss [82,83]. The migration and reaction of alkaline components with surrounding acidic chemicals might also result in mass loss and pore volume increase. All this results in the drop of compressive strength. Meanwhile, compared with specimens made using tap water or pure water, the mass and strength loss of seawater-mixed samples was lower, suggesting improved acid resistance [37]. This is also supported by the evidence that seawater-mixed AAM registered less medium capillary pores than its pure water and tap water counterparts, implying relatively less severe acid penetration (Figure 8). Seawater can enhance the alkalinity of the AAM matrix to some extent, delaying the degradation of obtained samples in acidic conditions.

Similar phenomena were found in terms of the mass and strength evolution of seawater AAM immersed in 3% MgSO_4_ solution. However, the mass and strength increase lasted up to 28 days, which is assigned to the formation of gypsum and ettringite, depositing into voids and pores [84]. Meanwhile, the expansion of ettringite, together with leaching of alkali and incorporation of sulfate into polymerized gels, caused negative effects on the microstructure and strength of resulting samples, leading to a mass and strength decrease [85,86]. Both gypsum and ettringite were detected after a 90-day exposure in H_2_SO_4_ and Mg_2_SO_4_ solutions. The gypsum might stem from the decomposition of polymerized gels, and the presence of ettringite can be due to the further reaction of gypsum with aluminum ions. Both phases can be further atmospherically carbonated, forming calcite. From the data measured using FT-IR and NMR, higher Si/Al ratio was detected in seawater-mixed samples when compared to tap water- and pure water-mixed samples after exposure. This might possibly be caused by the dealumination of the polymerized gels. Dealumination can result in an imperfect gel structure, consequently deteriorating the physical structure of AAMs. Meanwhile, Q4(1Al) peak was only recorded in a seawater AAM after sulfate exposure. The absence of this peak in pure water- and tap water-mixed samples implied the desilication process of zeolitic reaction products [87]. This kind of desilication can be related to the leaching of silicon [88], which also possesses effects on mechanical property of resulted AAMs.

### 4.2. Reinforcement Corrosion

Figure 9 summarizes the half-cell potential evolution of steel rebar in AAMs with seawater and sea sand. In Figure 9 (left), the AAM was produced in the one-part (just add water) way, using mixture of slag and fly ash as a precursor and solid sodium silicate, hydroxide and carbonate as an activator. Although control samples prepared using freshwater exhibited lower half-cell potential than OPC, the value (less than −0.35 V vs. CSE) in the seawater-mixed specimen was even lower. In ASTM C876, half-cell potential value more negative than −0.35 vs. CSE is related to corrosion probability of steel over 90%. The very low potential values in seawater AAM systems indicate that the corrosion risk of steel rebar is moderate to high [89,90,91,92]. However, after extracting the steel rebar after 1 year, only rare and scattered reddish stains were observed, demonstrating the corrosion took place but to a limited degree. The reinforcement from the control sample remained almost as it was originally, except adhering small fragments of pastes. The contradictory phenomenon can be attributed to sulfide components introduced by slag [92,93]. During alkali activation, much of the sulfur initially contained by slag can be released into the AAM matrixes [94,95]. HS^−^, S^2−^, SO_3_^2−^, S_2_O_3_^2−^ and SO_4_^2−^ were detected in the oxidation state of sulfur and accounted for 30–80% of the total sulfur in pore solution hydrated slag-blended cement [96,97,98,99]. Besides the highly reducing environment, the iron sulfides generated via reaction of sulfide anions with iron species can further passivate the steel surface and inhibits corrosion [96].The adoption of seawater might also contribute to create a reducing environment, since the high SO_4_^2−^ concentration can hinder the oxidation of S^2−^ and S^0^ [100].

Much higher potential values were recorded by Nguyen et al. [47] in fly ash-based geopolymer, using sea sand as fine aggregate. After exposing the samples to a Na_2_SO_4_ solution, the values of half-cell potential decreased considerably. The potential values were determined by strength grade, binder type and sulfate solution concentration. Enhancing geopolymer strength grade resulted in higher potential value at the same exposure conditions, while increasing sulfate concentration exhibited the contrary. In all cases, the steel rebar in a geopolymer matrix exhibited higher potential values than in the OPC mixture, suggesting that a geopolymer containing sea sand theoretically possesses better durability than its OPC counterpart.

It should be mentioned that all tests were conducted using the existing testing protocols for cement; the suitability is now open for discussion. Meanwhile, the findings from the abovementioned comparative investigation contribute to revealing the durability of AAMs.

## 5. Challenges and Perspectives

The adoption of seawater and sea sand in AAMs production possesses the advantages of reducing strain on freshwater and river sand, especially in some coastal and island areas with easy and rapid access to seawater and sea sand. Due to the limited data on this topic, it is hard to draw a conclusion as to whether the use of seawater and sea sand would compromise the engineering properties of obtained AAMs. In order to fill the knowledge gap and promote the sustainable development of construction materials, challenges in the design, modification, test and mechanism analysis of AAMs with feasible engineering properties and acceptable durability should be addressed.

Fundamental investigations on the reaction kinetics of seawater-mixed AAMs, revealing factors determining physicochemical properties ranging from setting time, slump, fluidity, etc., to long-term mechanical strength, chloride binding behavior, passivation and corrosion of reinforcement, etc., are essential to assess the AAMs containing seawater and sea sand and where these kinds of material can be used. To reveal the inherent characteristics of seawater/sea sand AAMs, extensive mix-design is proposed. Meanwhile, there are too many experimental variables that should be considered, and relationships among these variables are too complex. A combination of experimental investigation with computational techniques might be helpful to speed up the investigation process. For a convenient data interpretation and comprehensive understanding of the results from individual laboratories, identifying the physicochemical properties of source materials (AAMs binders, activators, seawater, sea sand, etc.), detailed mixture proportions, manufacture and characterization framework, etc., are recommended.

The risk of reinforcement corrosion is one of the main concerns related to the long-term safety of structures based on seawater/sea sand AAMs. The application of FRP in cement concrete has been demonstrated to achieve not only superior engineering performance prior to steel reinforcement, but also financial benefits in terms of low life-cycle cost [101,102]. This is also confirmed by the current findings from the C3 program (Carbon Concrete Composites) in Germany, in which no reduction caused by external influences seems currently possible for FRP reinforcement. Promising results have also been obtained in FRP-reinforced AAMs containing seawater and sea sand [58]. Beyond the corrosion resistance property, FRP is normally of high strength-to-weight ratio and able to adapt to various structural requirements, which might revolutionize structure construction.

## 6. Conclusions

Cementitious material is a crucial part of construction industry, but is associated with massive consumption of natural resources. The development of AAMs with seawater and sea sand achieves economic and ecological advantages, such as valorization of industrial wastes/byproducts, reduction of CO_2_ emission, and the alleviation of the freshwater shortage. In this work, the current progress in developing and assessing the chemical and engineering properties of AAMs with seawater and sea sand are reviewed. The effects of seawater on the workability and mechanical properties of AAMs vary, but a decrease in setting time and increase in strength were more frequently observed. The high salt content in seawater can accelerate the alkali activation process, resulting in the rapid formation of reaction products. In contrast, the large amount of salt on the sea sand’s surface can weaken the binding and interlocking between paste and aggregate. Thus, replacing river sand with sea sand led to a strength decrease. A contrary phenomenon was detected when coral sand was adopted, since its coarse surface provided good interlocking and its porous characteristics promoted internal curing.

The drying shrinkage of AAMs with sea sand at ambient conditions was higher, while the mass loss of seawater-mixed AAMs at 100 ℃ was decreased because of retarded water evaporation. Compared with OPC concrete of same strength grade, more pronounced aggregate breaking was observed in AAMs, and the bond failures of FRP in AAMs included pull-out and cracking or rupture. In terms of main reaction products, little difference has been reported after adopting seawater. Formation of Cl-bearing secondary products, such as Friedel’s salts, Cl-hydrocalumite and Cl-hydrotalcite, contributed to Cl-binding. The seawater-mixed AAMs showed mass and strength gain after short-term immersion in H_2_SO_4_ or MgSO_4_ solution, while scaling and deterioration were observed when extending the exposure time. No obvious reinforcement corrosion was observed in steel bar-reinforced seawater AAMs, although the half-cell potential values were very low. Efforts should be made to reveal the durability characteristics of AAMs with seawater and sea sand, and to verify the suitability of current testing protocols. Although some promising results have been reported, the systematic design, modification, testing and mechanism investigations of AAMs with seawater and sea sand are required to fill the knowledge gap and develop sustainable construction materials with feasible engineering properties and acceptable durability.

## Figures and Tables

**Figure 1 materials-16-03571-f001:**
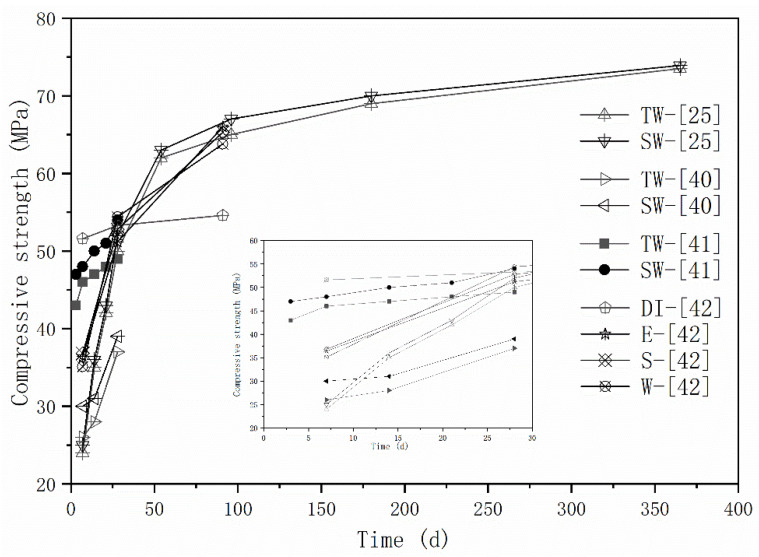
Strength evolution of AAMs with seawater and sea sand (E, S, W represents seawater sampled from different areas, adapted from [25,40,41,42]).

**Figure 2 materials-16-03571-f002:**
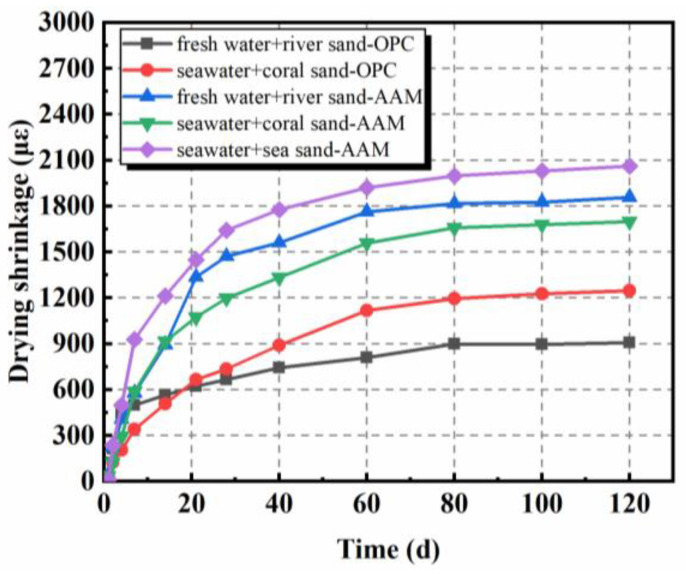
Drying shrinkage of AAMs with seawater and sea sand (adapted from [28,29]).

**Figure 3 materials-16-03571-f003:**
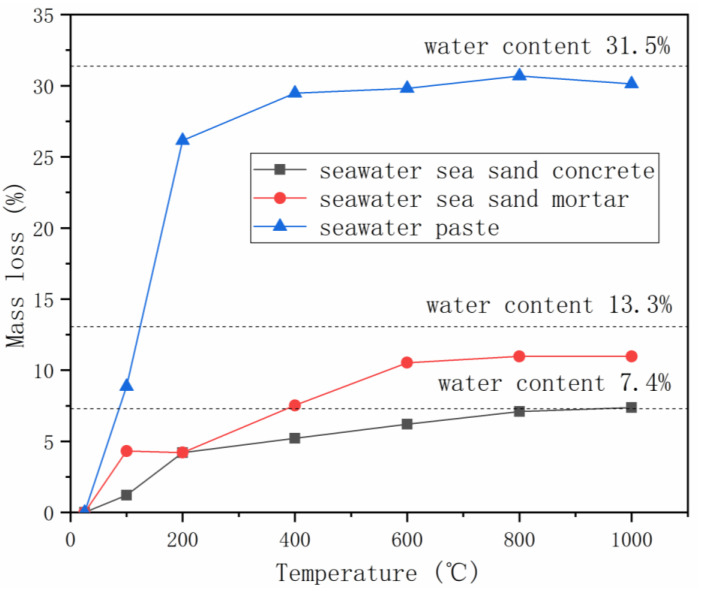
Mass loss of AAMs with seawater and sea sand at elevated temperatures (adapted from [38]).

**Figure 4 materials-16-03571-f004:**
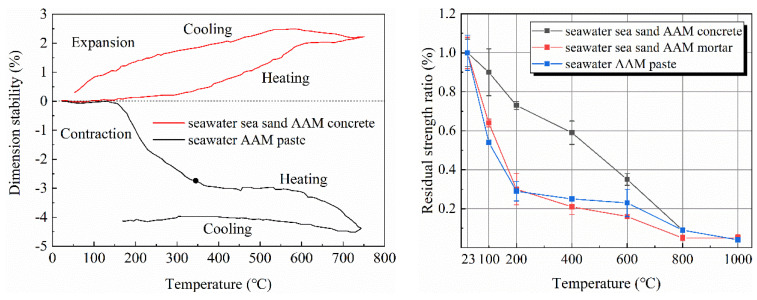
Dimension stability (**left**) and residual strength (**right**) of AAMs with seawater and sea sand at elevated temperature (adapted from [38]).

**Figure 5 materials-16-03571-f005:**
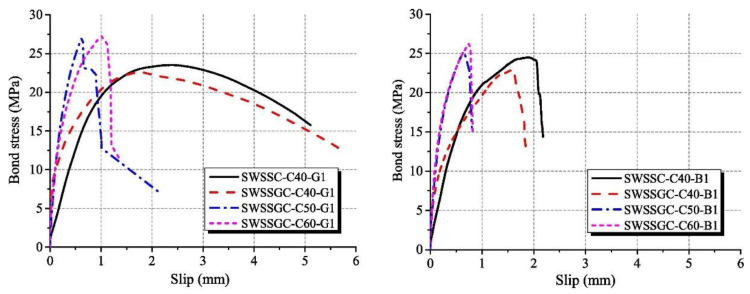
Bond stress-slip curves of AAMs concrete ([58]). (SWSSC: seawater sea sand OPC concrete; SWSSGC: seawater sea sand AAM concrete; C40/50/60: strength grade; G1: GFRB bars; B1: BFRB bars).

**Figure 6 materials-16-03571-f006:**
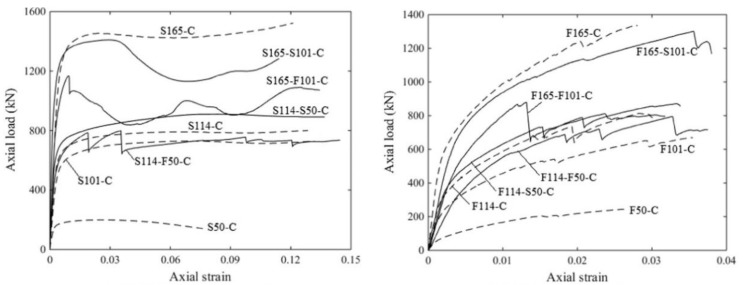
Load-strain curves of AAMs concrete filled tubes ([59]). (S: stainless steel tube; F: GFRP tube; 50/101/114/165: tube diameter in mm; C: AAM concrete filled).

**Figure 7 materials-16-03571-f007:**
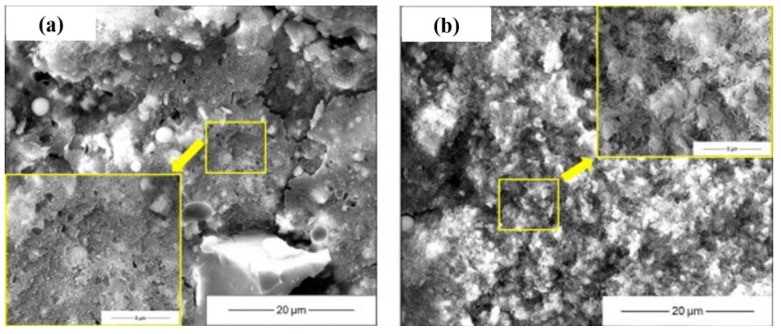
Microstructure of AAMs with seawater and sea sand ((**a**,**c**): tap water mixed AAM at 1 d and 28 d; (**b**,**d**): seawater-mixed AAM at 1 d and 28 d, adapted from [24]; (**e**,**f**): AAM mortar with sea sand and river sand, adapted from [47]).

**Figure 8 materials-16-03571-f008:**
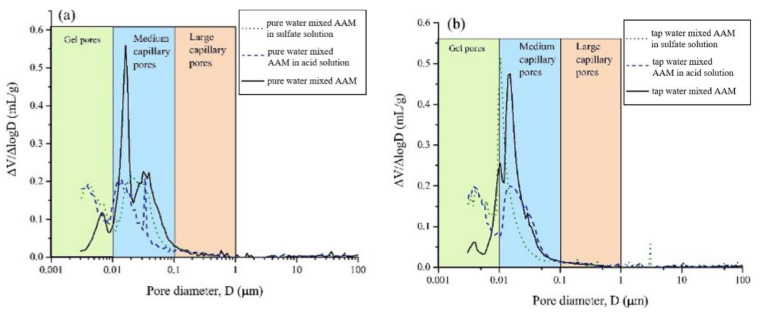
Pore size distribution of AAMs with pure water (**a**), tap water (**b**) and seawater (**c**) after exposure in acid and sulfate solution ([37]).

**Figure 9 materials-16-03571-f009:**
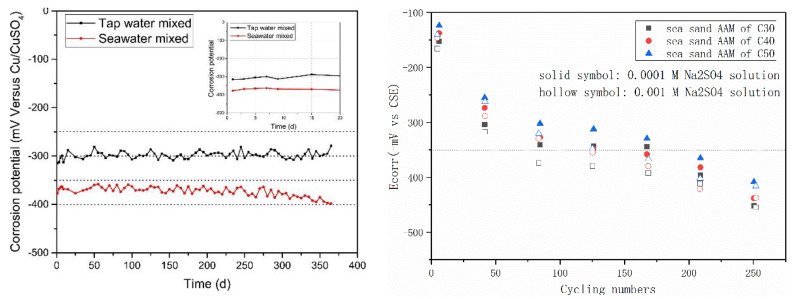
Half-cell potentials of steel rebar embedded in AAMs with seawater (**left**, [25]) and sea sand (**right**, [47]).

**Table 1 materials-16-03571-t001:** Setting time of AAMs with seawater and sea sand.

ID in Literature	Initial Setting Time (min)	Final Setting Time (min)	Slump (mm)	Mixture	Reference
Tw system	65	125	275	Precursor was a blend of calcium silicate slag, ground granulated blast furnace slag GGBS and fly ash (FA); tap water (Tw) or seawater (Sw) was mixed with sodium silicate to obtain Na_2_O/H_2_O mass ratio of 0.1. Water-to-blend ratio was 0.5.	[24]
Sw system	60	125	235		
Tap water-mixed	55	125	-	GGBS and FA as precursor; solid sodium hydroxide, silicate and carbonate were used as activator, water-(tap water or seawater)-to-solid ratio was 0.45.	[25]
Seawater-mixed	50	110	-		
Tap water	23	29	-	GGBS with anhydrous sodium metasilicate (SiO_2_/Na_2_O molar ratio 0.9) with mass ratio of 9:1 was used as solid source, and water-to-solid ratio was 1:2.9.	[26]
Sea water	31	53	-		
FR-OPC	186	336	183	Precursor is a mixture of GGBS, FA and SF, while seawater was mixed with sodium hydroxide and silicate as activator to obtain modulus of 1.2. Paste was of Na_2_O to binder mass ratio 4%, and mortar was of sand (river sand, sea sand, coral sand) to binder ratio 2.	[28]
SC-OPC	142	274	148		
FR-AAM	133	239	180		
SC-AAM	110	177	150		
SS-AAM	177	279	196		
TM1	122	223	168	Paste proportion was the same as above, but mixture of sea sand and coral sand (varying ratios) was used as aggregate.	[29]
TM2	113	213	168		
TM3	82	158	167		
TM4	72	106	170		
TM5	126	219	180		
TM6	133	210	196		
TM7	51	73	138		
TM8	53	74	144		
TM9	109	173	178		
TM10	85	133	156		
TM11	98	160	183		
TM12	53	74	140		
TM13	53	73	140		
TM14	40	60	143		
TM15	40	56	180		
TM16	32	48	178		

**Table 2 materials-16-03571-t002:** Chemical compositions of seawater from different regions (given in mg/L).

Cl^−^	Na^+^	SO_4_^2−^	Mg^2+^	Ca^2+^	K^+^	Region	Reference
3000	1800	410	240	98	67	Baltic	[30]
17,035	10,231	2800	1006	327	-	Nanisivik, Baffin Island	[31]
19,130	10,750	18,900	1370	320	380	-	[32]
22,330	11,400	3070	1328	422	399	Kish Island, Iran	[33]
19,400	9500	258	1100	350	350	Trondheim fjord	[34]
26,000	15,000	3700	2300	500	520	Arabian Gulf near UAE	[35]
18,980	10,556	2649	1262	400	380	-	[36]
41,942	7359	6802	1129	381.4	316.2	West Sea in Republic of Korea	[37]
20,700	11,940	3420	1430	439	622	Melbourne	[38]
22,100	8500	2600	1340	410	430	Songjeong, Republic of Korea	[39]
16,600	8300	2600	920	310	450	Samcheok, Republic of Korea	[40]
19,000	7500	3300	880	400	490	Geoje, Republic of Korea	[40]
16,000	7400	3000	890	390	470	Seochen, Republic of Korea	[40]

## Data Availability

Not applicable.

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
