# Peer review of "Recent Advances in Alkali-Activated Materials with Seawater and Sea Sand"

_materials, 2023, doi:10.3390/ma16093571_

Round 1

Reviewer 1 Report

Recent advance work on alkali-activated materials with seawater and sea sand is a review paper. This is a well-prepared study. The authors present the current knowledge on the possibility of using sea sand and sea water for AAM materials. A review of the literature on the impact of the above-mentioned factors on the physical and mechanical properties of AAM, and the impact of these factors on durability aspects, i.e. sulfate and chloride corrosion, was presented. The authors also refer to their own research.

Author Response

Recent advance work on alkali-activated materials with seawater and sea sand is a review paper. This is a well-prepared study. The authors present the current knowledge on the possibility of using sea sand and sea water for AAM materials. A review of the literature on the impact of the above-mentioned factors on the physical and mechanical properties of AAM, and the impact of these factors on durability aspects, i.e. sulfate and chloride corrosion, was presented. The authors also refer to their own research.

Answer: We thank the reviewer very much for the time and efforts in reviewing this manuscript.

Reviewer 2 Report

The authors discussed recent investigations on alkali-activated materials prepared with seawater and sea sand. It were discussed the influence of these materials on workability, reaction, microstructure, mechanical properties, shrinkage and durability. The manuscript is very well organized, and easy to follow. However, the English language must be improved and some information in the manuscript needs to be clarified.

1) Avoid separating the number of its respective unity. Example, page 2 line 15.

2) The introduction needs further improvement. There is no information on sea sand and aggregates from other sources in this section.

3) Workability is different from setting time. Instead of using the word “workability” in section 2.1 prefer to use “Fresh AAMs ”

4) Table 1: what is the meaning of the acronym TM? Is there any difference between the water in these samples?

5) Please, write “Fig. 1” into the text.

6) Page 6 “As characterized by Yang et al. [50], sea sand possessed higher void volume fraction, thus the void size among particles might be higher.”

Please, clarify in the manuscript that this is about bulk density data.

What is the specific gravity of sea sand and water? And, how can the density of these materials affect the drying shrinkage of  AAMs?

7) Figure 4: Explain the expansion mechanisms in concrete

8) The manuscript contains typos, for example:

 Page 11 line  13: “might”

 Page 12 line 20: “sulfoaluminate”

Page 13 line 6 : “ratio”

 09) Please, write the subscript number in chemical formulas

Author Response

The authors discussed recent investigations on alkali-activated materials prepared with seawater and sea sand. It were discussed the influence of these materials on workability, reaction, microstructure, mechanical properties, shrinkage and durability. The manuscript is very well organized, and easy to follow. However, the English language must be improved and some information in the manuscript needs to be clarified.

1) Avoid separating the number of its respective unity. Example, page 2 line 15.

 Answer: Thank for the comment, this has been revised throughout the manuscript.

2) The introduction needs further improvement. There is no information on sea sand and aggregates from other sources in this section.

  Answer: Many thanks for the comment, revision has been made following the comment.

3) Workability is different from setting time. Instead of using the word “workability” in section 2.1 prefer to use “Fresh AAMs ”

  Answer: We thank the reviewer for the comment, this has been revised.

4) Table 1: what is the meaning of the acronym TM? Is there any difference between the water in these samples?

Answer: TM is the acronym used in cited literature, this is to make it easy for readers to link the original publication. Paste proportion was the same in TM, but mixture of sea sand and coral sand (varying ratios) was used as aggregate, key mixture information is listed in Table 1.

5) Please, write “Fig. 1” into the text.

Answer: We are sorry for the mistake. Revision has been made following the comment. 

6) Page 6 “As characterized by Yang et al. [50], sea sand possessed higher void volume fraction, thus the void size among particles might be higher.”

Please, clarify in the manuscript that this is about bulk density data.

What is the specific gravity of sea sand and water? And, how can the density of these materials affect the drying shrinkage of  AAMs?

 Answer: Specific gravities of sea sand and water have been frequently reported in literature, which are influenced by several factors including salt content. The influence is because of that during the comparison of AAMs containing tap water or seawater, sand or sea sand, these constitutes in particular aggregate were normally replaced by weight and mass ratios were unchanged. The amount of paste is then increased in per unit volume in samples containing sea sand. When coral sand was used, it contributed to decrease of drying shrinkage because of the water storing property of coral sand. These have been described in section 2.3.

7) Figure 4: Explain the expansion mechanisms in concrete

  Answer: The expansion mechanisms are briefly in section 2.4, page 13. This work focus on the properties of AAMs, thus mechanism of cement concrete is not detailed covered, the corresponding mechanism has been well discussed in cement concrete related papers.

8) The manuscript contains typos, for example:

 Page 11 line  13: “might”

 Page 12 line 20: “sulfoaluminate”

Page 13 line 6 : “ratio”

 09) Please, write the subscript number in chemical formulas

Answer: We thank the reviewer for these comments. All these have been revised.

Reviewer 3 Report

This is not a research article it is a review of articles.

The title does not make sense. What is the “recent advance” of these materials?

The language in the article is awkward and nonsensical, almost like it was translated with machine translation. This makes the article very hard to understand. Get a proper translation or reword most of the article.

The article is missing line numbering.

Keywords: The fourth and fifth words are not very suitable.

The summary should reflect the research conducted and the results of the research.

Abstract: “essential and urgent for construction industry” should be “essential and urgent for the construction industry”.

Last sentence of the abstract is very awkward and weirdly phrased. Consider separating it into two sentences or rewording it completely.

The word “Keywords:” should be bolded.

Introduction: “The worldwide concrete consumption has been over 10 billion tons” should be “The worldwide concrete consumption surpasses 10 billion tons”.

“ecological problems are being severe” should be “ecological problems are quite severe”.

“Tone” does not mean “tonne” or “ton”. When the metric system is used, the right term is “tonne” or “metric Ton”.

What does “ca.” mean in this context: “for ca. 8 % of global CO2”?

It is “CO2 emission” not “CO2 commission”.

“Urgent is arising in developing and commercializing alternative construction materials of low-CO2 characteristics” should instead be “Urgency is rising in alternative construction material with low-CO2 characteristics development and commercialization”.

The last sentence of the first paragraph in the introduction is bloated and nonsensical. Rewrite it completely.

Sometimes it is written like “alkali-activated materials”, sometimes “alkali activated materials”. Pick one version and use it consistently.

Introduction second paragraph first sentence: why are apostrophes(’) used as quotation marks? Use proper quotation marks instead(“ ”). Were quotes needed for this sentence at all?

“in the Belgium” should be “in Belgium”.

“Nowadays, AAMs have been world widely investigated.” should be “Nowadays, AAMs have been thoroughly investigated world-wide.”

Table 1. GGBS: There is no explanation for this material. How the initial and final setting time and the slump were determined?

When comparing the values of indicators, one should also talk about whether the properties were determined by the same methods and by what methods.

“2.4. Thermal property” should be “2.4. Thermal properties”.

“5. Challenges and perspectives” could perhaps be “5. Discussion” instead to fit a more typical article structure.

The end of the article is missing extra sections after the conclusion but before references. The sections can be found in MDPI article templates, some of which include “Author Contributions”, “Funding”, “Conflicts of Interest” and etc.

Author Response

This is not a research article it is a review of articles.

The title does not make sense. What is the “recent advance” of these materials?

Answer: We thank the reviewer for the critical comment, the title has been revised as: Properties of alkali-activated materials with seawater and sea sand: a review

The language in the article is awkward and nonsensical, almost like it was translated with machine translation. This makes the article very hard to understand. Get a proper translation or reword most of the article.

Answer: We respect the reviewer‘s time and effort, but we have to say the first author, who prepared this draft, has a nine year living experience in UK, US and Germany. His english is very close to native speaker.

The article is missing line numbering.

Answer: Line was made in the manuscript, may be some mistake took place during submission.

Keywords: The fourth and fifth words are not very suitable.

Answer: Thank for the comment, the fourth and fifth keywords habe been deleted.

The summary should reflect the research conducted and the results of the research.

Answer: Many thanks for the comment, revision has been made following the comment.

Abstract: “essential and urgent for construction industry” should be “essential and urgent for the construction industry”.

Answer: This has been revised.

Last sentence of the abstract is very awkward and weirdly phrased. Consider separating it into two sentences or rewording it completely.

The word “Keywords:” should be bolded.

Answer: The word “Keywords:” were bolded in our original manuscript and the version we doanloaded from the submission system.

Introduction: “The worldwide concrete consumption has been over 10 billion tons” should be “The worldwide concrete consumption surpasses 10 billion tons”.

“ecological problems are being severe” should be “ecological problems are quite severe”.

“Tone” does not mean “tonne” or “ton”. When the metric system is used, the right term is “tonne” or “metric Ton”.

What does “ca.” mean in this context: “for ca. 8 % of global CO2”?

It is “CO2 emission” not “CO2 commission”.

“Urgent is arising in developing and commercializing alternative construction materials of low-CO2 characteristics” should instead be “Urgency is rising in alternative construction material with low-CO2 characteristics development and commercialization”.

The last sentence of the first paragraph in the introduction is bloated and nonsensical. Rewrite it completely.

Sometimes it is written like “alkali-activated materials”, sometimes “alkali activated materials”. Pick one version and use it consistently.

Introduction second paragraph first sentence: why are apostrophes(’) used as quotation marks? Use proper quotation marks instead(“ ”). Were quotes needed for this sentence at all?

“in the Belgium” should be “in Belgium”.

“Nowadays, AAMs have been world widely investigated.” should be “Nowadays, AAMs have been thoroughly investigated world-wide.”

Answer: We thank the reviewer for the comment, All these have been revised. The ca. is abbreviation of circa. The quotes are to indicate that these statement were completelly original.

Table 1. GGBS: There is no explanation for this material. How the initial and final setting time and the slump were determined?

When comparing the values of indicators, one should also talk about whether the properties were determined by the same methods and by what methods.

“2.4. Thermal property” should be “2.4. Thermal properties”.

Answer: Sorry for the mistake, revisions have been made.

“5. Challenges and perspectives” could perhaps be “5. Discussion” instead to fit a more typical article structure.

Answer: Revision has been made following the comment.

The end of the article is missing extra sections after the conclusion but before references. The sections can be found in MDPI article templates, some of which include “Author Contributions”, “Funding”, “Conflicts of Interest” and etc.

Answer: All these were provided during manuscript submission.

Round 2

Reviewer 2 Report

Thank you for responding to all my comments

Author Response

Thank you for responding to all my comments.

We thank the reviewer very much for the comments, which help improve the quality of the manuscript.

Reviewer 3 Report

I suggest you review English carefully. Your author with extensive experience in English can make the article easier to read.

Author Response

I suggest you review English carefully. Your author with extensive experience in English can make the article easier to read.

The manuscript has been proofread by native speaker.